# The ESSnuSB Design Study: Overview and Future Prospects

A. Alekou [1,2], E. Baussan [3], A. K. Bhattacharyya [4], N. Blaskovic Kraljevic [4], M. Blennow [5,6], M. Bogomilov [7], B. Bolling [4], E. Bouquerel [3], F. Bramati [8], A. Branca [8], O. Buchan [4], A. Burgman [9], C. J. Carlile [2], J. Cederkall [9], S. Choubey [5,6], P. Christiansen [9], M. Collins [4,10], E. Cristaldo Morales [8], L. D'Alessi [3], H. Danared [4], D. Dancila [2], J. P. A. M. de André [3], J. P. Delahaye [1], M. Dracos [3], I. Efthymiopoulos [1], T. Ekelöf [2], M. Eshraqi [4,10], G. Fanourakis [11], A. Farricker [12], E. Fernandez-Martinez [13], B. Folsom [4], T. Fukuda [14], N. Gazis [4], B. Gålnander [15], Th. Geralis [11], M. Ghosh [16], A. Giarnetti [17], G. Gokbulut [18,†], L. Halić [16], M. Jenssen [4], R. Johansson [4], A. Kayis Topaksu [18], B. Kildetoft [4], B. Kliček [16,*], M. Kozioł [19], K. Krhač [16,‡], Ł. Łacny [20], M. Lindroos [4], A. Longhin [21,22], C. Maiano [4], S. Marangoni [8], C. Marrelli [4], C. Martins [4], D. Meloni [17], M. Mezzetto [22], N. Milas [4], M. Oglakci [18], T. Ohlsson [5,6], M. Olvegård [2], T. Ota [13], M. Pari [21,22], J. Park [9,§], D. Patrzalek [4], G. Petkov [7], P. Poussot [3], F. Pupilli [22], S. Rosauro-Alcaraz [23], D. Saiang [24], J. Snamina [19], A. Sosa [4], G. Stavropoulos [11], M. Stipčević [16], B. Szybiński [20], R. Tarkeshian [4], F. Terranova [8], J. Thomas [3], T. Tolba [25], E. Trachanas [4], R. Tsenov [7], G. Vankova-Kirilova [7], N. Vassilopoulos [26], E. Wildner [1], J. Wurtz [3], O. Zormpa [11] and Y. Zou [2]

1    CERN, 1211 Geneva 23, Switzerland
2    Department of Physics and Astronomy, FREIA Division, Uppsala University, P.O. Box 516, 751 20 Uppsala, Sweden
3    IPHC, Université de Strasbourg, CNRS/IN2P3, F-67037 Strasbourg, France
4    European Spallation Source, P.O. Box 176, SE-221 00 Lund, Sweden
5    Department of Physics, School of Engineering Sciences, KTH Royal Institute of Technology, Roslagstullsbacken 21, 106 91 Stockholm, Sweden
6    The Oskar Klein Centre, AlbaNova University Center, Roslagstullsbacken 21, 106 91 Stockholm, Sweden
7    Faculty of Physics, Sofia University St. Kliment Ohridski, 1164 Sofia, Bulgaria
8    University of Milano-Bicocca and INFN Sez. di Milano-Bicocca, 20126 Milano, Italy
9    Department of Physics, Lund University, P.O. Box 118, 221 00 Lund, Sweden
10   Faculty of Engineering, Lund University, P.O. Box 118, 221 00 Lund, Sweden
11   Institute of Nuclear and Particle Physics, NCSR Demokritos, Neapoleos 27, 15341 Agia Paraskevi, Greece
12   Cockroft Institute (A36), Liverpool University, Warrington WA4 4AD, UK
13   Departamento de Fisica Teorica and Instituto de Fisica Teorica, IFT-UAM/CSIC, Universidad Autonoma de Madrid, Cantoblanco, 28049 Madrid, Spain
14   Institute for Advanced Research, Nagoya University, Nagoya 464-8601, Japan
15   GSI Helmholtzzentrum für Schwerionenforschung GmbH Planckstraße 1, 64291 Darmstadt, Germany
16   Center of Excellence for Advanced Materials and Sensing Devices, Ruđer Bošković Institute, 10000 Zagreb, Croatia
17   Dipartimento di Matematica e Fisica, Universitá di Roma Tre, Via della Vasca Navale 84, 00146 Rome, Italy
18   Department of Physics, Faculty of Science and Letters, University of Cukurova, 01330 Adana, Turkey
19   AGH University of Science and Technology, al. Mickiewicza 30, 30-059 Krakow, Poland
20   Faculty of Mechanical Engineering, Cracow University of Technology, Al. Jana Pawła II 37, 31-864 Krakow, Poland
21   Department of Physics and Astronomy "G. Galilei", University of Padova, Via Marzolo 8, 35131 Padova, Italy
22   INFN Sezione di Padova, Via Marzolo 8, 35131 Padova, Italy
23   Pôle Théorie, Laboratoire de Physique des 2 Infinis Iréne Joliot Curie (UMR 9012) CNRS/IN2P3, 15 Rue Georges Clemenceau, 91400 Orsay, France
24   Department of Civil, Environmental and Natural Resources Engineering, Luleå University of Technology, SE-971 87 Luleå, Sweden
25   Institute for Experimental Physics, Hamburg University, 22761 Hamburg, Germany
26   Institute of High Energy Physics (IHEP) Dongguan Campus, Chinese Academy of Sciences (CAS), 1 Zhongziyuan Road, Dongguan 523803, China
*    Correspondence: budimir.klicek@irb.hr
†    Current address: Department of Physics and Astronomy, Ghent University, Proeftuinstraat 86, 9000 Ghent, Belgium.
‡    Current address: Department of Applied Mathematics, University of Twente, 7500 AE Enschede, The Netherlands.
§    Current address: Center for Exotic Nuclear Studies, Institute for Basic Science, Daejeon 34126, Republic of Korea.

**Abstract:** ESSnuSB is a design study for an experiment to measure the CP violation in the leptonic sector at the second neutrino oscillation maximum using a neutrino beam driven by the uniquely powerful ESS linear accelerator. The reduced impact of systematic errors on sensitivity at the second maximum allows for a very precise measurement of the CP violating parameter. This review describes the fundamental advantages of measurement at the second maximum, the necessary upgrades to the ESS linac in order to produce a neutrino beam, the near and far detector complexes, and the expected physics reach of the proposed ESSnuSB experiment, concluding with the near future developments aimed at the project realization.

**Keywords:** neutrino; oscillation; long baseline; CP violation; second maximum; precision

## 1. Introduction

It was widely believed in the first half of the 20th century that all physical laws must be invariant to spatial translation (implying the conservation of momentum), spatial rotation (implying the conservation of angular momentum), time translation (implying the conservation of energy), parity transformation, and time inversion. Parity transformation effectively maps our world to its mirror image: the invariance of physics to parity transformation means that for any physical process there exists a mirrored version of it. Note that parity transformation formally maps a vector to its negative value, i.e., $\vec{x} \to -\vec{x}$, while mirroring inverts only one of its Cartesian coordinates, e.g., $(x, y, z) \to (-x, y, z)$; mirroring can be obtained by applying parity transformation followed by a rotation, which makes the two transformations equivalent assuming rotational invariance. By the 1950s, parity invariance was experimentally verified with a high degree of precision for the strong and electromagnetic interactions, but there was no conclusive evidence for weak interactions [1]. A landmark experiment was then performed by observing beta decay of polarized cobalt-60 nuclei, which found that parity is in fact maximally violated in that process [2].

Shortly thereafter it was proposed that the true symmetry of the Universe is actually charge-parity (CP) symmetry [3], where charge transformation C maps each particle to its antiparticle and vice versa. That is, laws of physics remain invariant to the parity inversion if one also swaps all particles and antiparticles. The CP symmetry violation (CPV) would then imply a fundamental difference in behavior between particles and antiparticles.

The CPV has been observed in the hadron sector, first by measuring decays of neutral kaons [4] and later by measuring decays of heavier neutral mesons [5–8]. The size of the CPV in the hadron sector has turned out to be quite small, not enough to explain the matter–antimatter asymmetry that we observe in the Universe today [9,10].

At the time of writing, there is no conclusive evidence of CPV in the lepton sector; there are hints from the T2K [11–13] and NO$\nu$A [14] experiments, though, that leptonic CP might actually be maximally violated. The sensitivity of these two experiments is not expected to reach the discovery level of $5\,\sigma$. Therefore, next generation facilities need to be constructed, which will have larger target mass and more intense neutrino beams required to collect a statistical sample significant enough to resolve the existence of the leptonic CP violation. Two such experiments are currently in the construction phase: the Hyper-Kamiokande (HyperK) [15] and the Deep Underground Neutrino Experiment (DUNE) [16–19]. They are expected to have the ability to reject the no-CPV hypothesis with a significance of more than $5\,\sigma$ for a large fraction of parameter space [20–24].

ESSnuSB [25,26] will be a next-to-next generation CPV experiment focusing on the precise measurement of CPV parameters in the lepton sector. The unprecedented precision will be achieved by measuring neutrino oscillations in the second oscillation maximum—in which the CPV effect is about 2.7 times larger than in the first one—making the experiment less sensitive to the systematic errors. This will be made possible using a very intense neutrino beam produced by the uniquely powerful ESS linear accelerator together with the very large neutrino far detectors.

This review starts with the discussion on fundamental benefits of measurement at the second neutrino oscillation maximum. It proceeds with the description of the proposed ESSnuSB experiment: neutrino beam production, near and far neutrino detectors, and the physics reach of the proposed experiment. It concludes with a brief description of the future developments towards the realization of the project.

## 2. CP Violation Measurement at the Second Oscillation Maximum

At a fundamental level, ESSnuSB aims to measure the CPV by observing the difference in oscillation probabilities between neutrinos and antineutrinos in $\nu_\mu \to \nu_e$ and $\overline{\nu}_\mu \to \overline{\nu}_e$ appearance channels, respectively.

### 2.1. Oscillations in Vacuum

The probability of neutrino oscillations in vacuum assuming a plane–wave approximation is given by (see Section 14.4 in [27])

$$P_{\nu_\alpha \to \nu_\beta} = \delta_{\alpha\beta} - 4 \sum_{i>j} \mathrm{Re}\left(A_{ij}^{\alpha\beta}\right) \sin^2 \frac{\Delta m_{ij}^2 L}{4E} \pm 2 \sum_{i>j} \mathrm{Im}\left(A_{ij}^{\alpha\beta}\right) \sin \frac{\Delta m_{ij}^2 L}{2E} , \tag{1}$$

where $\alpha$ is the initial neutrino flavor, $\beta$ is the oscillated flavor, indices $i$, and $j$ are in the range 1–3, $A_{ij}^{\alpha\beta} = U_{\alpha i}^* U_{\alpha j} U_{\beta i} U_{\beta j}^*$ is a quadrilinear product of elements of the unitary PMNS [28–32] mixing matrix $U$, $\Delta m_{ij}^2 = m_i^2 - m_j^2$ is a difference of squared neutrino masses $m_i$ and $m_j$, $E$ is the energy of the neutrino and $L$ is the distance between neutrino creation and interaction points; the upper sign $(+)$ in front of the third term corresponds to neutrinos, the lower one $(-)$ to antineutrinos.

The difference between oscillation probabilities of neutrinos and antineutrinos is then given by the expression

$$\mathcal{A}_{\mathrm{CP}}^{\alpha \to \beta} = P_{\nu_\alpha \to \nu_\beta} - P_{\overline{\nu}_\alpha \to \overline{\nu}_\beta} = 4 \sum_{i>j} \mathrm{Im}\left(A_{ij}^{\alpha\beta}\right) \sin \frac{\Delta m_{ij}^2 L}{2E} . \tag{2}$$

It follows from the properties of the three-generation mixing matrix that the term $\mathrm{Im}\left(A_{ij}^{\alpha\beta}\right)$ is constant up to a sign if $i \neq j$ and $\alpha \neq \beta$, and zero otherwise [33]. That is,

$$\mathrm{Im}\left(A_{ij}^{\alpha\beta}\right) \equiv \pm J , \tag{3}$$

where $J$ is called the Jarlskog invariant.

Assuming the commonly used parametrization (see Section 14.3 in [27]) of the mixing matrix

$$U = \begin{pmatrix} c_{12}c_{13} & s_{12}c_{13} & s_{13}e^{-i\delta_{\mathrm{CP}}} \\ -s_{12}c_{23} - c_{12}s_{23}s_{13}e^{i\delta_{\mathrm{CP}}} & c_{12}c_{23} - s_{12}s_{23}s_{13}e^{i\delta_{\mathrm{CP}}} & s_{23}c_{13} \\ s_{12}s_{23} - c_{12}c_{23}s_{13}e^{i\delta_{\mathrm{CP}}} & -c_{12}s_{23} - s_{12}c_{23}s_{13}e^{i\delta_{\mathrm{CP}}} & c_{23}c_{13} \end{pmatrix} , \tag{4}$$

where $c_{ij} = \cos \theta_{ij}$ and $s_{ij} = \sin \theta_{ij}$ are sine and cosine of a mixing angle $\theta_{ij}$, and $\delta_{\mathrm{CP}}$ is a CPV phase, the Jarlskog invariant can be written as

$$J = s_{12}c_{12}s_{13}c_{13}s_{23}c_{23}c_{13} \sin \delta_{\mathrm{CP}} . \tag{5}$$

The CPV amplitude for the ESSnuSB oscillation channel can then be written as

$$\mathcal{A}_{\mathrm{CP}}^{\mu \to e} = P_{\nu_\mu \to \nu_e} - P_{\overline{\nu}_\mu \to \overline{\nu}_e} = -16J \sin \frac{\Delta m_{31}^2 L}{4E} \sin \frac{\Delta m_{32}^2 L}{4E} \sin \frac{\Delta m_{21}^2 L}{4E} . \tag{6}$$

As an illustration, the dependence of $\mathcal{A}_{\text{CP}}^{\mu\to e}$ on $L/E$, using central values of parameters from Table 1 and $\delta_{\text{CP}} = -\pi/2$, is shown in Figure 1.

By denoting $x_{\max}^{(1)}$ and $x_{\max}^{(2)}$ as the values of $L/E$ at the first and second maximum, respectively, one obtains the expression for the ratio between CPV violation at the second and first maximum:

$$\frac{\mathcal{A}_{\text{CP}}^{\mu\to e}\left(x_{\max}^{(2)}\right)}{\mathcal{A}_{\text{CP}}^{\mu\to e}\left(x_{\max}^{(1)}\right)} = \frac{\sin\frac{\Delta m_{31}^2 x_{\max}^{(2)}}{4}\,\sin\frac{\Delta m_{32}^2 x_{\max}^{(2)}}{4}\,\sin\frac{\Delta m_{21}^2 x_{\max}^{(2)}}{4}}{\sin\frac{\Delta m_{31}^2 x_{\max}^{(1)}}{4}\,\sin\frac{\Delta m_{32}^2 x_{\max}^{(1)}}{4}\,\sin\frac{\Delta m_{21}^2 x_{\max}^{(1)}}{4}} \,. \tag{7}$$

The ratio between CPV in second and first maximum does not depend on the PMNS mixing angles, only on the neutrino mass splittings. Plugging in the values for mass splittings from Table 1, together with $x_{\max}^{(1)}$ and $x_{\max}^{(2)}$, one obtains

$$\frac{\mathcal{A}_{\text{CP}}^{\mu\to e}\left(x_{\max}^{(2)}\right)}{\mathcal{A}_{\text{CP}}^{\mu\to e}\left(x_{\max}^{(1)}\right)} \approx 2.7 \,. \tag{8}$$

That is, the difference between neutrino and antineutrino oscillation probabilities due to CP violation is about three times larger at the second maximum than at the first one.

**Table 1.** The best-fit values and $1\,\sigma$ allowed regions of the oscillation parameters used throughout this review, as given in [34]. Reprinted from [25,26].

| Parameter | Best-Fit Value $\pm 1\sigma$ Range |
|---|---|
| $\sin^2\theta_{12}$ | $0.304 \pm 0.012$ |
| $\sin^2\theta_{13}$ | $0.02246 \pm 0.00062$ |
| $\sin^2 2\theta_{23}$ | $0.9898 \pm 0.0077$ |
| $\Delta m_{21}^2$ | $(7.42 \pm 0.21) \times 10^{-5}\ \text{eV}^2$ |
| $\Delta m_{31}^2$ | $(2.510 \pm 0.027) \times 10^{-3}\ \text{eV}^2$ |

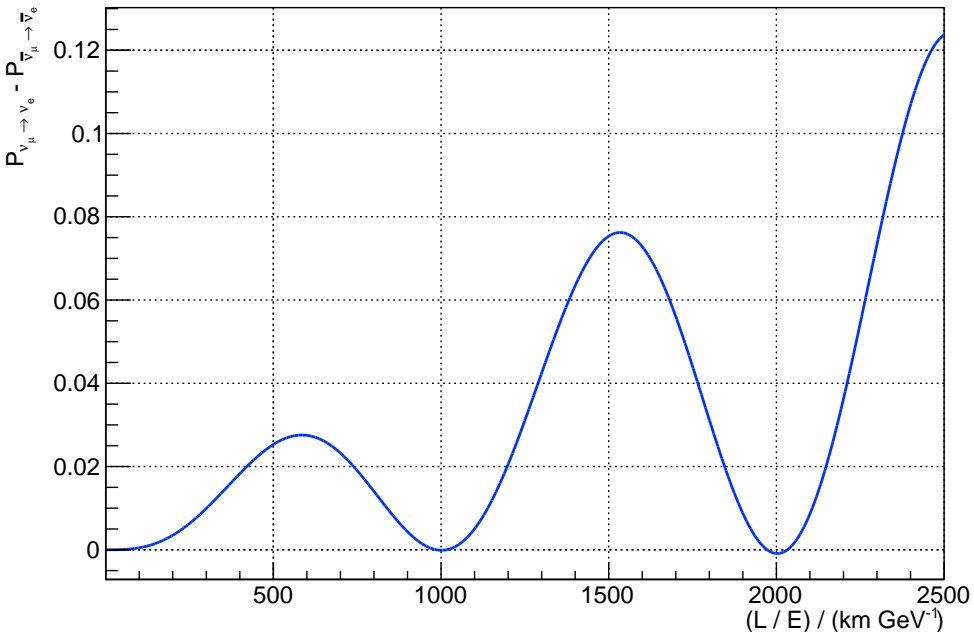

**Figure 1.** Dependence of $\mathcal{A}_{\text{CP}}^{\mu\to e}$ on ratio $L/E$. The first maximum of the CPV amplitude is situated at $L/E \approx 585$ km/GeV and the second at $L/E \approx 1534$ km/GeV.

### 2.2. Matter Effects

When neutrinos propagate through matter instead of vacuum, the oscillation probability functions change. This effect is well understood and described in the literature [35–38]. The fundamental reason for this is the effective potential induced by forward elastic scattering of neutrinos with matter. The matter potential seen by electron neutrinos is different than the one seen by muon and tau neutrinos: forward elastic scattering can proceed through neutral-current (NC) interactions for all three neutrino flavors, while there is an additional contribution for $\nu_e$ only through charged-current (CC) interactions with orbital electrons.

Matter effects may mimic the vacuum CPV signal (one may argue that introducing matter, and not antimatter, into vacuum breaks the CP symmetry in itself). This needs to be carefully taken into account in the long baseline oscillation experiments since there neutrino beam propagates through the Earth's crust. This effect is illustrated in Figure 2 for the distance of 360 km at which the far detector of the ESSnuSB experiment is going to be located.

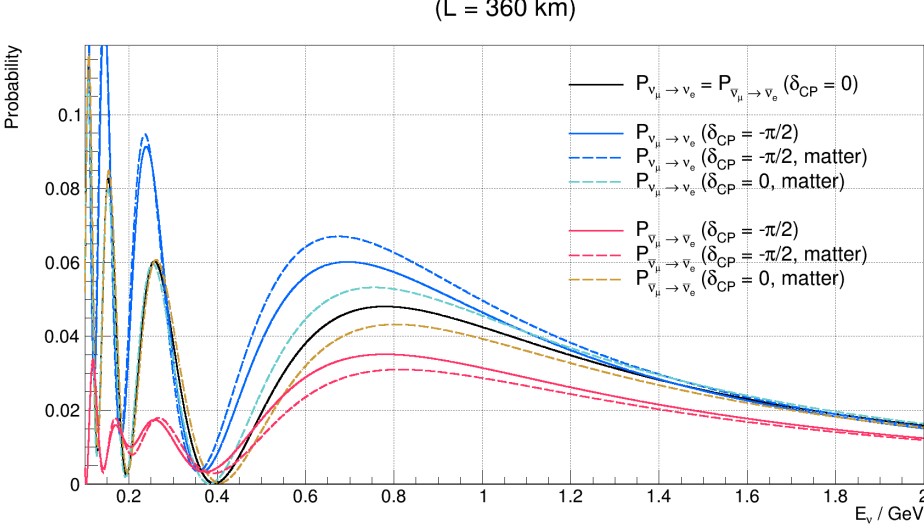

**Figure 2.** $\nu_\mu \rightarrow \nu_e$ and $\overline{\nu}_\mu \rightarrow \overline{\nu}_e$ oscillation probabilities as a function of neutrino energy at the fixed distance of 360 km. The oscillation probabilities are shown for $\delta_{CP} = 0$ and $\delta_{CP} = -\pi/2$. Full lines correspond to oscillations in vacuum and dashed lines to oscillations in matter.

The first appearance maxima in Figure 2 are located roughly in the neutrino energy region of 0.65–0.85 MeV and the second in the region of 0.25–0.35 MeV. The exact position of a particular maximum depends on the choice of value for $\delta_{CP}$ and whether the matter effects are taken into account. The oscillation probability is significantly altered in the presence of matter around the first maximum, while at the second maximum the probabilities in matter and in vacuum are similar. In fact, it can be shown that at neutrino energies of currently operating and proposed long baseline experiments and terrestrial matter densities, the matter effects around the second maximum have minimal contribution to probability functions [39,40].

### 2.3. Summary

The advantages of measurement at the second oscillation maximum over the first one are significantly larger difference between neutrino and antineutrino oscillation probabilities, and minimal dependance of oscillation functions on matter effects. The downside is that the second maximum is about three times further away from the source than the first maximum for a given neutrino energy, implying a reduction of neutrino flux—consequently the number of expected neutrino interactions—by a factor of nine. Alternatively, one could move from the first to the second oscillation maximum by keeping the distance and lowering the neutrino energy which would also result in the lower interaction rate due to smaller

neutrino interaction cross-section and larger dispersion of the beam; this is unfeasible in the ESSnuSB setup because one would quickly go below the $\nu_\mu$ CC interaction threshold by lowering the already low neutrino energy. However, given the larger CPV amplitude (8) and assuming that the background and systematic error are comparable at the first and second maximum, the measurement at the second is expected to be more significant and precise if large enough statistical sample can be accumulated [22,41–47]. To accumulate a significant statistical sample at the second maximum, a very intense neutrino beam is required. To achieve this, the ESSnuSB project foresees to use the uniquely powerful ESS proton linear accelerator currently in construction near Lund in Sweden.

### 3. Neutrino Beam

Neutrino beam for the ESSnuSB experiment will be produced using the ESS [48] proton linear accelerator.

The basic idea behind the neutrino beam production is to create a beam of pions which is allowed to decay in-flight via the process $\pi^+ \rightarrow \mu^+ + \nu_\mu$ (and its charge conjugate). The pions are produced by shooting a proton beam onto a thin target: this produces a number of hadronic species which immediately decay via strong and electromagnetic interactions, leaving only weak-stable particles called secondaries. Secondaries are composed of pions and heavier hadrons (mostly mesons like kaons, strange and charmed mesons), the number of latter rising with the proton beam energy. An electromagnetic horn envelops the target and is used to simultaneously focus the particles of a selected charge sign and defocus those of the opposite sign. By selecting the sign of the focused particles, one can choose between producing a beam of neutrinos and a beam of antineutrinos—decays of positive mesons mostly produce neutrinos, while decays of negative ones mostly produce antineutrinos. While charged pion decays produce almost exclusively a muon (anti)neutrino beam, heavier mesons (such as kaons) have decay channels that include electron (anti)neutrinos which contribute to the $\nu_e$ and/or $\overline{\nu}_e$ component of the produced beam. Decays of heavier mesons such as $D_s$ may have tau neutrinos in the final state as well, but their energy production threshold is well above the ESS proton energies.

Since CPV measurement will be performed by observing electron (anti)neutrino interactions from $\nu_\mu \rightarrow \nu_e$ and $\overline{\nu}_\mu \rightarrow \overline{\nu}_e$ oscillation channels, the prompt $\nu_e$ and $\overline{\nu}_e$ beam components constitute the background to this measurement. The electron neutrino component of the beam coming from decays of heavier mesons is difficult to model precisely due to the quantum chromodynamical (QCD) nature of meson production, which induces an additional systematic error on the CPV measurement. An advantage of using a relatively low energy proton beam such as ESS at 2.5 GeV is that secondaries will contain a very small amount of flavored hadrons due to their mass production threshold, which in turn makes the resulting muon (anti)neutrino beam quite clean.

Since the ESS accelerator was designed for the production of spallation neutrons, a number of modifications will be required to enable production of a neutrino beam in parallel with the neutron program. The proposed changes are shown in Figure 3.

The ESS will operate using 2.86 ms long proton (or $H^-$ ion) pulses, which would produce neutrino beam pulses of a comparable duration. It has been shown that such "long" pulses cannot be used for CPV measurement (see Section 6.3.4 in [26]) due to the atmospheric neutrino background at the ESSnuSB far detectors: the number of atmospheric neutrino interactions during the "long" pulse would be so high that its statistical fluctuations would be larger than the number of expected beam $\nu_e$ interactions, completely drowning the CPV signal. To solve this problem, the ESS pulses will be compressed to 1.3 μs which will effectively eliminate the atmospheric background in the far detectors.

The proton accumulator ring of 384 m in length will be used to compress the ESS pulses. This will be performed by filling the accumulator ring over many of its periods of circulation and then discharging it towards the neutrino targets in one period, creating a 1.3 μs proton pulse. In order to efficiently fill the ring, $H^-$ ions will be accelerated by the ESS instead of protons, their two extra electrons will be stripped at the moment they

enter the accumulator ring: this will avoid space charge issues which would arise from the electromagnetic repulsion between the protons already circulating within the ring and those being injected. The long ESS H⁻ pulse will be chopped into four subpieces in the low energy part of the ESS linac, and each of the subpieces will be compressed to 1.3 µs one after another. Each of the four compressed pulses will be transferred to a separate neutrino target using the switchyard system located between the accumulator ring and the target station.

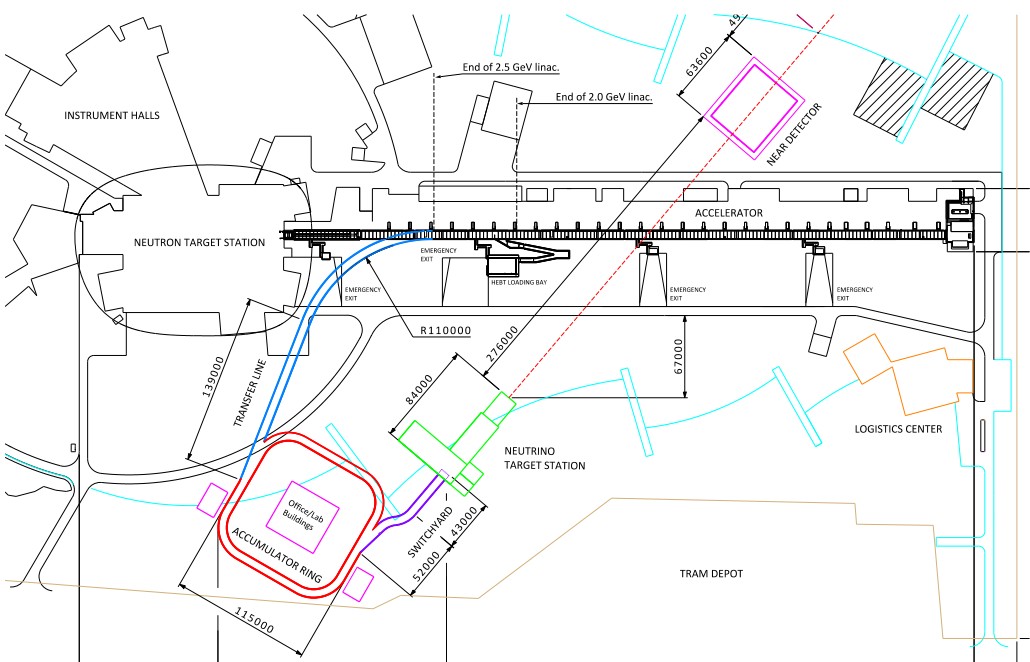

**Figure 3.** Layout of the ESS accelerator including ESSnuSB modifications required for neutrino beam production and detection. The proposed modifications are show in color: the transfer line (blue), the accumulatior ring (red), the swithcyard (indigo), the neutrino target station (green), and the near detector hall (purple). Reprinted from [25,26].

In order to withstand the very high power of the ESS linac, the ESSnuSB target station will consist of four identical target/horn systems, each receiving a quarter (1.25 MW) of the nominal average 5 MW ESS beam power. The targets will have a cylindrical shape of 78 cm in length and 3 cm in diameter and will consist of closely packed titanium spheres of 3 mm mean diameter cooled by gaseous helium. The secondaries will be focused using a pulsed magnetic horn driven by 350 kA pulses of 100 µs duration and will be led into a 50 m long decay tunnel. The charge sign of the focused secondaries is determined by the direction of the electric current through the horn: one can switch between the positive charge focusing (neutrino mode) and negative charge focusing (antineutrino mode) by reversing the current. The shape of the horn and length of the decay tunnel have been optimized for maximum CPV measurement sensitivity using a genetic algorithm.

The neutrino beam energy distribution using this setup is shown in Figure 4. The neutrino flux is dominated by $\nu_\mu(\overline{\nu}_\mu)$ component in neutrino(antineutrino) mode, which makes up a 97.6 % (94.8%) of the flux. The $\overline{\nu}_\mu(\nu_\mu)$ component in neutrino(antineutrino) mode makes up 1.7% (4.7%) and comes from wrong-sign pion decays (i.e., imperfect charge selection of secondaries) and from decays of tertiary muons (muons produced in the secondary pion decay). The $\nu_e(\overline{\nu}_e)$ component in neutrino(antineutrino) mode makes up 0.67% (0.43%) of the flux and comes dominantly from the tertiary muon decays, with a small component from three-body kaon decay. The very small $\overline{\nu}_e(\nu_e)$ in neutrino(antineutrino) mode, making up a 0.03% (0.03%) of the flux, comes primarily from wrong-sign tertiary muon decays.

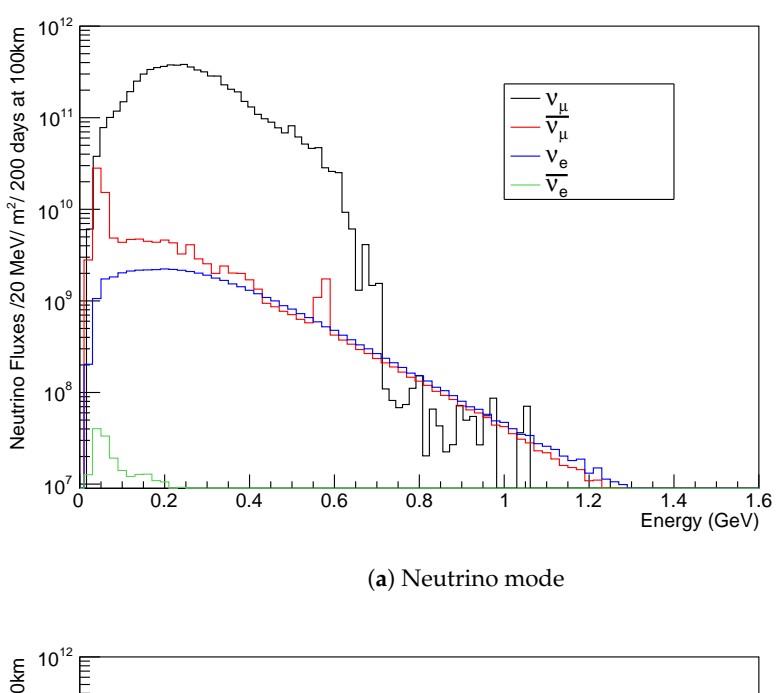

(**a**) Neutrino mode

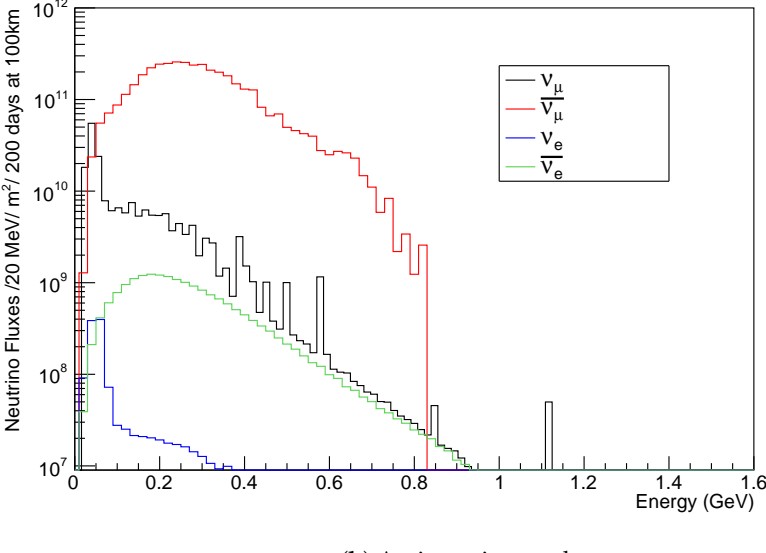

(**b**) Antineutrino mode

**Figure 4.** The expected neutrino flux components and their energy distribution at the 100 km distance from the source, in absence of the neutrino oscillations. Reprinted from [25,26].

It should be noted that modeling tertiary muon production is much more precise than modeling secondary kaon production. This, together with the fact that most of the electron (anti)neutrino component comes from tertiary muon decays, makes it possible to model this component with less systematic uncertainty than would be the case for more energetic proton beams which produce a larger number of kaon secondaries.

## 4. Neutrino Detectors

The ESSnuSB experiment will consist of a suite of near detectors and two very large water Cherenkov (WC) far detectors. The far detectors will be used to measure the oscillated neutrino spectrum and hence the signal for the CP-violation. Measurements from the near detectors will be used to constrain the prompt neutrino flux and interaction cross-sections. A detailed description of the ESSnuSB detectors can be found in the ESSnuSB conceptual design report [25,26]; this section provides a brief overview of the basic ideas.

### 4.1. Near Detectors

Knowledge of neutrino interaction cross-sections, both inclusive and differential, will play a crucial role in the ESSnuSB experiment. An a priori uncertainty of the cross-sections directly translates to uncertainty of the expected event rate at the far detectors, which in turn decreases the CPV discovery potential and precision of the $\delta_{CP}$ phase measurement. It turns out that this uncertainty (assuming it stays within reasonable bounds) does not affect the $5\,\sigma\;\delta_{CP}$ coverage too much (see Figure 11)—mostly due to the very intense neutrino beam, large far detectors and the fact that we are measuring at the second oscillation maximum. However, it does play a crucial role in the expected precision of the $\delta_{CP}$ measurement (see Figure 9). Since ESSnuSB is designed to be a next-to-next generation CPV experiment with focus on precision, near detectors have been designed with a focus on cross-section measurement. Additionally, separate cross-section measurement campaigns are foreseen in the construction and commissioning phase of the ESSnuSB experiment (see Section 6).

The near detector suite of the ESSnuSB project will consist of three different detectors, in upstream to downstream order: the emulsion detector with water as a target, the SFGD-like granulated scintillator detector and the water Cherenkov detector. The near detector hall is schematically shown in Figure 5.

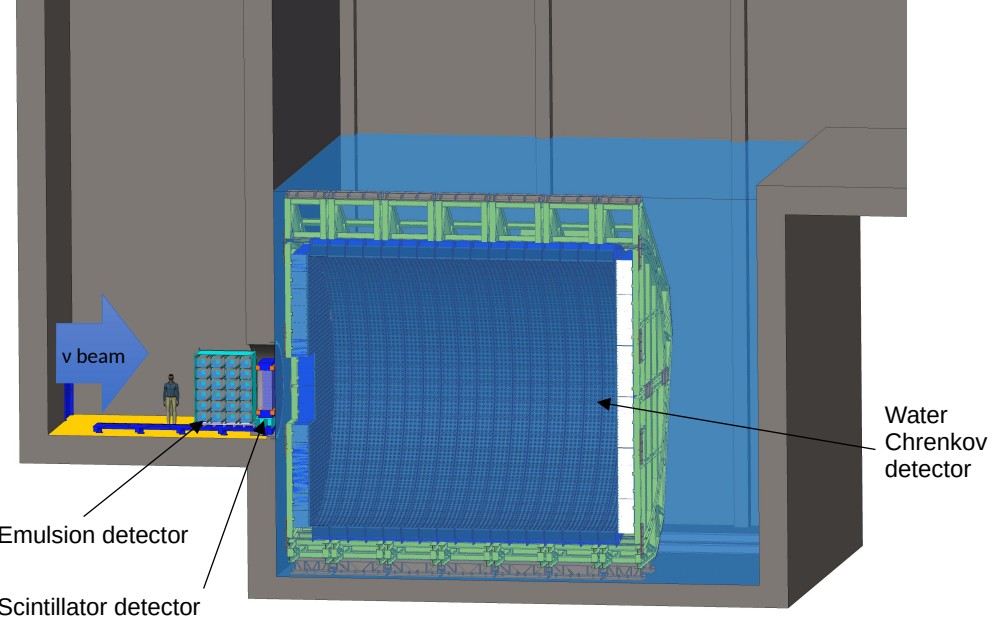

**Figure 5.** Schematic view of the ESSnuSB near detector hall.

The emulsion near detector (named $\nu$iking) will have a 1 t water target mass and will be used to precisely measure the final state topology of the neutrino–water interactions. This will make it possible to discriminate between different neutrino interaction modes—e.g., quasi-elastic scattering (QES), inelastic scattering such as resonant (RES) and deep-inelastic (DIS) scattering, and meson exchange current (MEC) scattering. The measurement of the MEC contribution to the total cross-section will be of special importance since it is expected to have a significant contribution at ESSnuSB neutrino energies and yet this channel is currently not adequately explored. The signature of this interaction is a charged lepton and two nucleons in the final state—at least one of the two is a proton in neutrino interactions, a neutron in anti-neutrino interactions. Usage of emulsion technology will enable a direct measurement of the proton component in the final state, which will make it possible to precisely tag and measure the kinematics of these interactions.

Immediately downstream of the emulsion detector, a SFGD-like [49] magnetized granulated scintillator detector will be installed. It will feature a fiducial target mass of 1 t and a dipole magnetic field of up to 1 T perpendicular to the beam direction. Having a magnetic

field, it will be able to discriminate between positively and negatively charged leptons and therefore between neutrinos and antineutrinos. Its granular design will enable both muon momentum measurement and calorimetric measurement of the final state particles in neutrino interactions. Hence, it will feature the best neutrino energy reconstruction of the three near detectors. The downside is that the target material will be composed of hydrocarbons (CH) instead of water; a theoretical model can be used to go from the CH to water neutrino interaction cross-section, but this will introduce additional systematic uncertainties. The additional purposes of this detector will be to *(i)* provide timing information to the emulsion detector and *(ii)* provide muon charge and momentum information to the near WC detector for those events in which the charged lepton in the final state transverses through both detectors—this will allow additional calibration of the WC detector.

Downstream of the scintillator detector, a 0.75 kt target mass ($\sim$420 kt fiducial) water Cherenkov detector will be situated. Due to its large mass, it is expected to record the bulk of the neutrino interactions among the three detectors. It will be used to collect a high-statistic sample of $\nu_\mu$ interactions and a significant sample of $\nu_e$ interactions. Additionally, a sample of neutrino–orbital electron scattering events $\nu + e^- \rightarrow \nu + e^-$ will be isolated. The interaction cross-section for this process is precisely known, so it can be used to directly measure the neutrino flux. This measurement, in turn, will be an input to a measurement of the neutrino-nucleus cross-section—the main goal of the near detector setup. It should be noted that having the same target material and detection technology as the far detector, it will be possible to correlate systematic uncertainties to some extent between the two detectors using a dedicated analysis.

*4.2. Far Detectors*

The far detector site will consist of two identical large water Cherenkov detectors. Each detector will be placed in a cavern in the shape of a standing cylinder with the height of 78 m and the base diameter of 78 m, having an extra room on top for access and housing of the required infrastructure (see Figure **??**). The design with two caverns was chosen due to the extreme technical challenge of excavating a single one large enough to contain the required water volume. A cylindrical structure 76 m high and having a 76 m base diameter will be constructed in the cavern to house the photomultipliers (PMT). The entire cavern will be filled with ultra-pure water. Assuming a 2 m fiducial cut inward from the walls of the PMT structure, each detector will contain a 270 kt fiducial mass of water, for a total of 540 kt fiducial mass.

The PMT-holding structure will feature inward-pointing 20-inch PMTs whose purpose will be to detect Cherenkov light from charged particles produced in neutrino interactions, and outward-facing 8-inch PMTs that will be used as a veto. The inner detector will have PMT coverage (fraction of the area covered by PMTs) of 30%.

In the ESSnuSB energy range (see Figure 4), most of the CC neutrino interactions are expected to be of the QES type. Since the only final state particle above the Cherenkov threshold in this type of interaction is a charged lepton (electron for $\nu_e$ and muon for $\nu_\mu$), high purity $\nu_\mu$ and $\nu_e$ event samples can be isolated with high efficiency; the selection algorithm does not need to achieve high purity and efficiency in the higher energy region containing a significant contribution from RES and DIS with complicated final states for the simple reason that not many ESSnuSB neutrino interactions will happen there. The resulting selection efficiency as a function of energy for different neutrino flavors is shown in Figure 6.

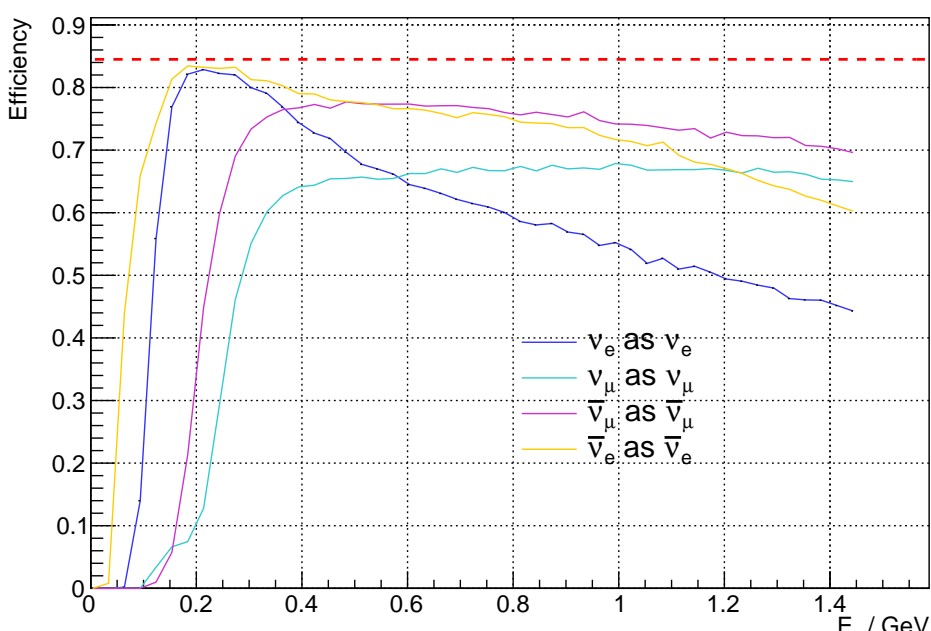

**Figure 6.** The efficiency to correctly select a neutrino flavor as a function of neutrino energy. Full lines correspond to different neutrino flavors. The dashed line is the efficiency of the fiducial cut. Reprinted from [25,26].

## 5. Physics Reach

The expected significance of the leptonic CPV discovery and the precision of $\delta_{CP}$ measurement has been studied (see Section 8 in [26]) assuming the described ESSnuSB setup. The total run-time is assumed to be 10 y, out of which 5 y will be in neutrino mode and 5 y in antineutrino mode. The expected measured neutrino spectra in the far detectors are shown in Figure 7. The fact that the measurement will be conducted in the $L/E$ range covering both second and first oscillation maxima results in the very high expected discovery potential and unprecedented precision of the CPV phase $\delta_{CP}$ measurement. These results have been shown to be very robust to different assumptions on the systematic uncertainties.

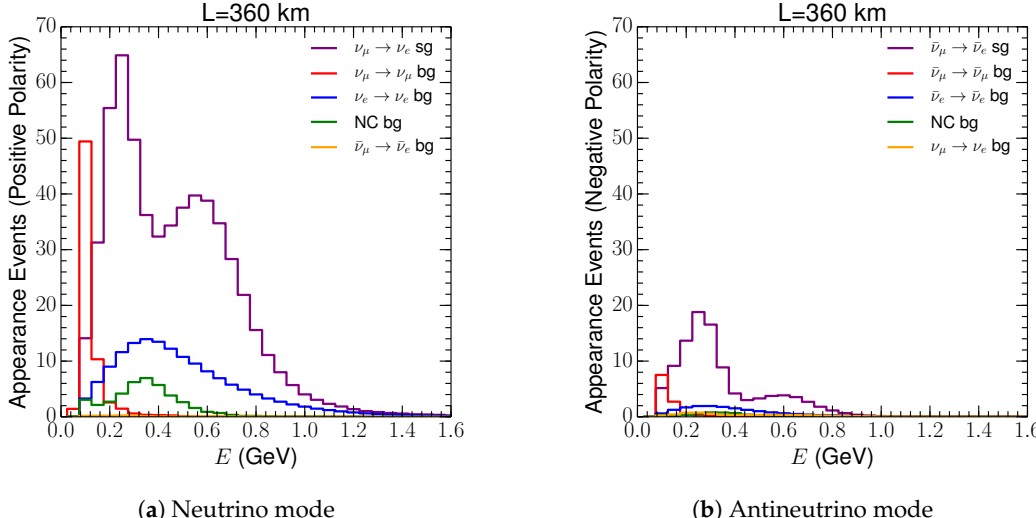

(**a**) Neutrino mode                                    (**b**) Antineutrino mode

**Figure 7.** The expected number of observed neutrino events as a function of reconstructed neutrino energy in the far detectors, shown for the signal channel and the most significant background channels. Each plot corresponds to 200 days (effective year) of data taking. Reprinted from [25,26].

The effect of three types of systematic error have been studied, each of them distorting the expected measured neutrino spectrum. These are *(i)* normalization uncertainty, uncertainty on the overall normalization of the expected neutrino spectrum; *(ii)* energy calibration uncertainty, assuming fully correlated error on neutrino energy reconstruction; and *(iii)* bin-to-bin uncorrelated uncertainty, modeling the distortion of the shape of the expected spectrum. All these systematics have been applied independently for each measurement channel; a single measurement channel corresponds to an oscillation channel such as $\nu_\mu \to \nu_e$ and $\nu_e \to \nu_e$, or to the NC neutrino interactions.

The CPV discovery potential is defined as the expected significance with which the experiment will rule out the CP-conserving values $\delta_{CP} = 0, \pi$. Out of the three studied systematic error types, it turns out that the CPV discovery potential is most sensitive to the normalization uncertainty. This effect is shown in Figure 8.

The extreme robustness of the ESSnuSB experiment to the systematic error is demonstrated by the fact that the CPV discovery potential is competitive even with a very pessimistic assumption on the normalization error of 25%. Unless otherwise noted, throughout the remainder of the text a smaller—but still conservative—normalization error of 5% will be used. The coverage of the $\delta_{CP}$ range for which the discovery potential is more than $5\sigma$ as a function of run-time is shown in Figure 9.

The strength of the ESSnuSB experiment will come from the unprecedented precision of the $\delta_{CP}$ measurement. The bin-to-bin uncorrelated systematic error has the largest effect on the $\delta_{CP}$ precision of the three. The $1\sigma$ precision as a function of true value of $\delta_{CP}$ is shown in Figure 10.

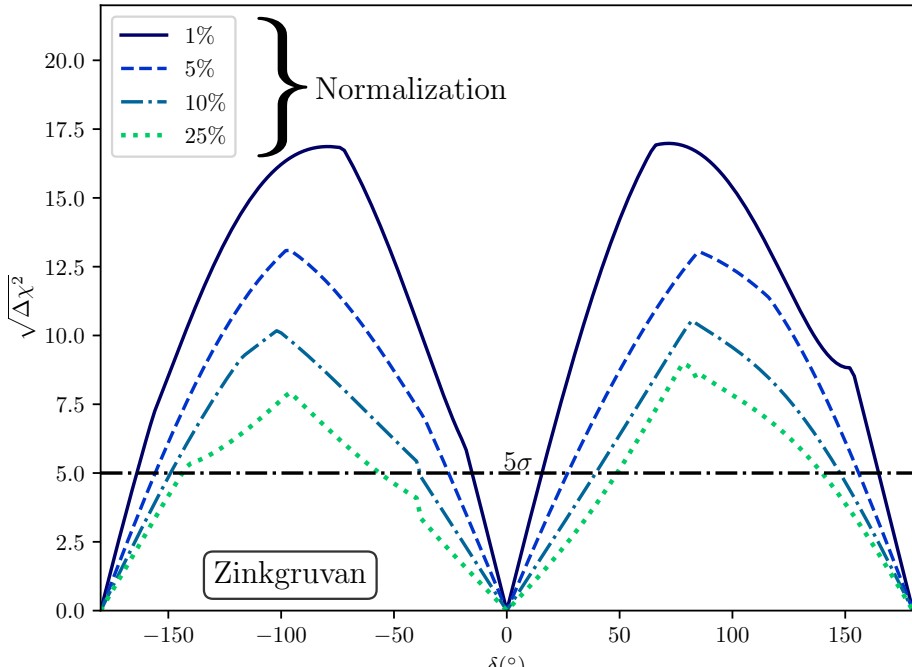

**Figure 8.** CPV discovery potential as a function of true $\delta_{CP}$ value, assuming the baseline of 360 km (Zinkgruvan mine) and run-time of 5 y in $\nu$ mode and 5 y in $\overline{\nu}$ mode. Different lines correspond to different normalization uncertainty assumptions. Reprinted from [25,26].

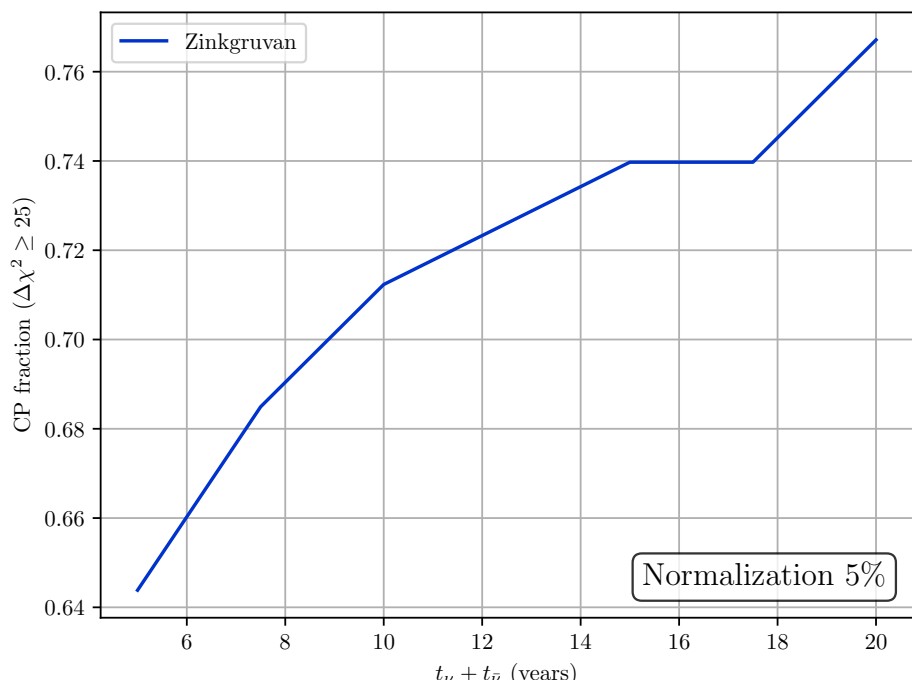

**Figure 9.** Coverage of the $\delta_{CP}$ range for which the discovery potential is larger than 5 $\sigma$ as a function of run-time, assuming equal time in neutrino mode and antineutrino mode.

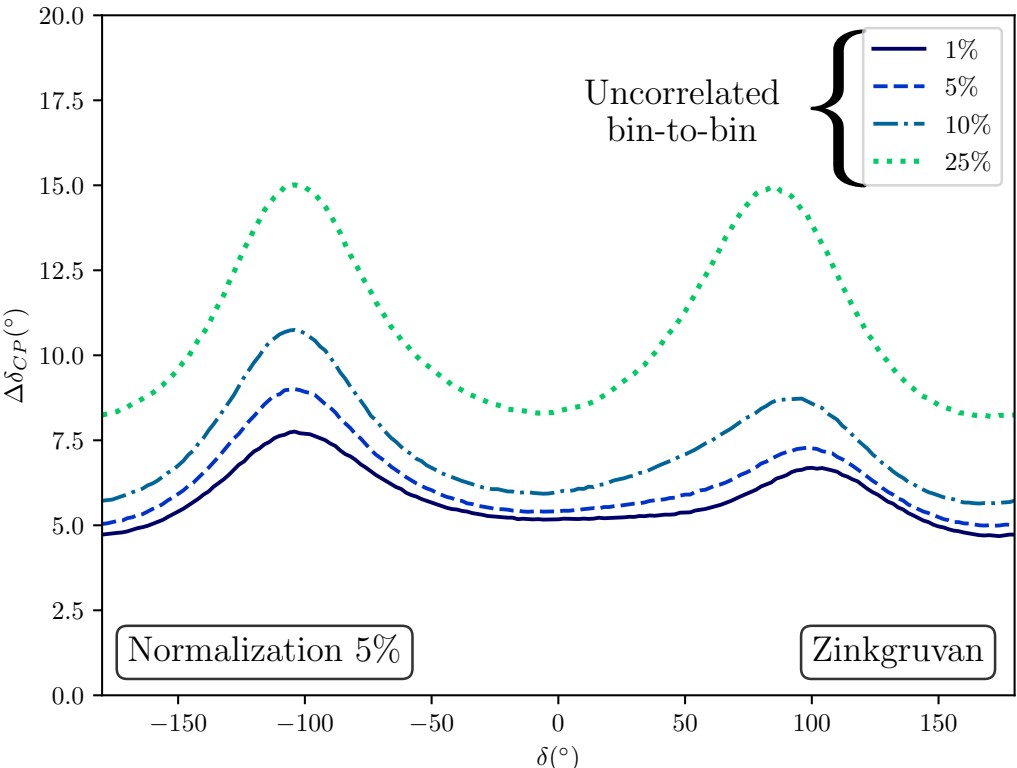

**Figure 10.** The expected 1 $\sigma$ precision for the measurement of the CPV parameter $\delta_{CP}$ as a function of the true value of $\delta_{CP}$, assuming the baseline of 360 km (Zinkgruvan mine) and run-time of 5 y in $\nu$ mode and 5 y in $\bar{\nu}$ mode. Different lines correspond to different bin-to-bin uncorrelated errors. A normalization error of 5% is applied on top of the bin-to-bin error. Reprinted from [25,26].

It can be seen from Figure 10 that the precision for $\delta_{CP}$ value is expected to be less than 9° for all true values when using a conservative assumption on the systematic uncertainty of 5% for normalization and 5% for bin-to-bin uncorrelated. The actual systematic uncertainties are expected to be smaller than that by the start of the project, further improving the precision. If the value of $\delta_{CP}$ is roughly known, the precision can be improved even further by adjusting the time measurements will be taken in neutrino mode vs. time in antineutrino mode while keeping the total run-time constant.

It is expected that by the time the ESSnuSB starts taking data, the existence of CPV in the lepton sector will be either confirmed or excluded in a large part of parameter space by the HyperK [15] and/or DUNE [16–19] experiments. In either case, ESSnuSB will be an important next-to-next generation experiment which will be able to precisely measure the amplitude of CPV or to conduct a more precise scan of the parameter space in search of its existence.

It should be noted that, apart from the CPV measurement, ESSnuSB will have an ability to discriminate between the normal and inverted neutrino mass hierarchy with a significance of more than 5 $\sigma$ [50].

## 6. Future Developments

The study performed in the ESSnuSB CDR [25,26] and described so far in this review has mainly focused on the CPV measurement using the ESS neutrino beam. The construction of the large far detector facility will require a significant amount of time and resources: an intermediate step is therefore foreseen, which will focus on the neutrino interaction cross-section measurement, neutrino production target station R&D and study of the additional physics potential of ESSnuSB near and far detectors. These topics will be studied within the ESSnuSB+ [51] project, together with the study of the civil engineering of both ESS upgrades and the far detector site and production of conceptual CAD drawings

of the infrastructure. The additional facilities at the ESS site proposed by the ESSnuSB+ project are shown in Figure 11.

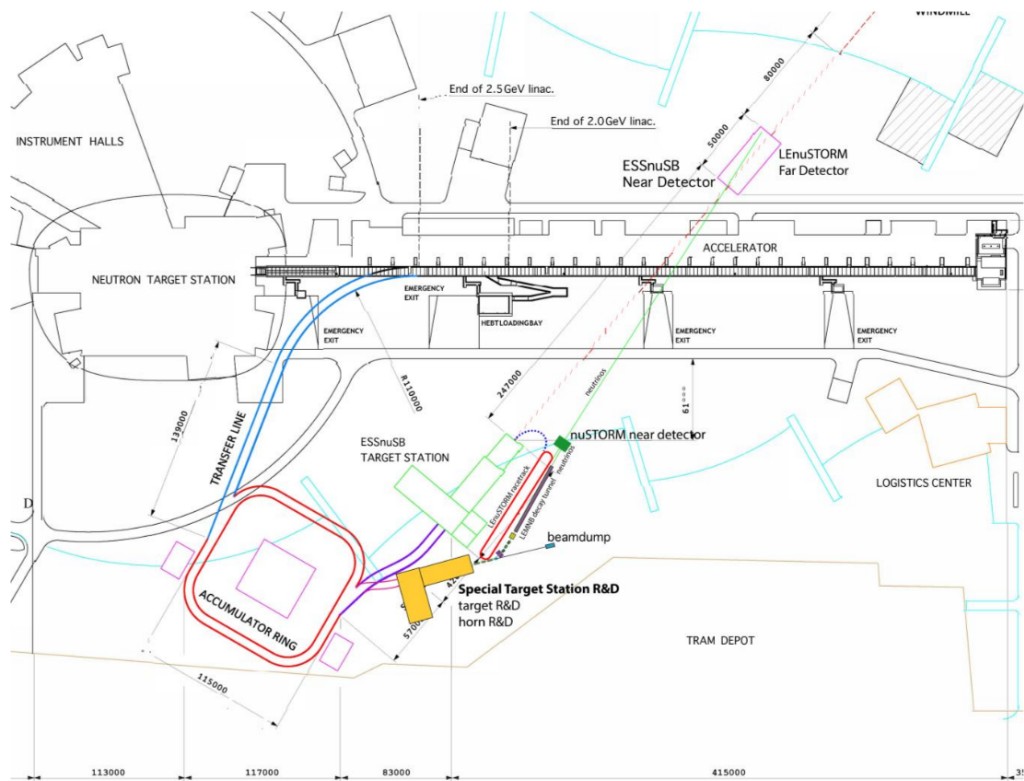

**Figure 11.** Layout of the proposed upgrades to the ESS linear accelerator, including those from ESSnuSB and ESSnuSB+. The ESSnuSB+ upgrades include a special target station, a muon storage racetrack ring (LEnuSTORM), a low-energy monitored neutrino beam (LEMNB) line, and a new near detector to be used both for LEnuSTORM and LEMNB. The ESSnuSB near detector will be used as a far detector for LEnuSTORM and LEMNB. The image has been reprinted from the ESSnuSB+ proposal.

As described in Section 3, the full ESSnuSB neutrino production target station will consist of four identical target/horn systems due to the high power of the ESS beam. The ESSnuSB+ proposes to build an R&D target station which will contain only one ESSnuSB target/horn operating at 1.25 MW beam power, i.e., at 1/4 of the full ESS power. The expertise obtained in design, construction and operation of this target station will directly apply to running a full-power ESSnuSB neutrino beam production facility. In addition, the pions from the secondary hadron beam produced in the R&D target station will be used to feed the low energy muon storage ring (LEnuSTORM), similar to the nuSTORM project [52].

A transfer line will be designed for the secondary particles—composed of hadrons (nucleons, pions and a small fraction of kaons) and their decay products composed mostly of muons and neutrinos—exiting the R&D target station; the pions will be led into a straight part of the LEnuSTORM racetrack ring in which pions will decay through the process $\pi^+ \to \mu^+ + \nu_\mu$ (or its charged conjugate version for $\pi^-$ mode of operation) to produce additional muons. The bend at the end of the straight section will be designed to kinematically select muons to keep them in the ring (coming from the decay of pions, muons will have a lower average momentum, making the kinematical selection possible even though the two particles have similar masses), while pions will be lead into the beam dump. The circulating muons will gradually decay through the process $\mu^+ \to e^+ + \nu_e + \bar{\nu}_\mu$ (or its charged conjugate version for $\pi^-/\mu^-$ mode); decays occurring in the straight section will produce a neutrino beam containing equal parts muon and electron neutrinos (with additional muon neutrinos coming from the pion decay in the first straight portion of the

ring during the filling). This beam will have a significant $\nu_e$ (or $\bar{\nu}_e$) component which will be used to measure the electron (anti)neutrino interaction cross-section.

A low energy monitored neutrino beam line, inspired by the ENUBET project [53], will be situated parallel to the LEnuSTORM ring. The basic idea behind this facility is to have an instrumented decay tunnel in which pions and muons decay to produce a neutrino beam. The walls of the decay tunnel will be instrumented with an iron-scintillator calorimeter which will be used to reconstruct the energy and direction of the charged decay products of pions and muons (muons for pion decay and electrons for muon decay). This information will be used to constrain the expected energy spectrum of neutrinos exiting the tunnel. The neutrinos will be detected in a detector shared with the LEnuSTORM. This will make it possible to precisely measure the interaction cross-section of muon neutrinos and possibly electron neutrinos. Given the very high expected number of decays in the tunnel, the LEMNB prefers to have a proton pulse as long as possible. Therefore, it will operate directly using "long" ESS pulses, bypassing the accumulator ring. This will require static focusing of the secondaries since electromagnetic horns are not able to withstand the long high-current pulses required to hold the magnetic field for 3 ms. The feasibility of such static focusing has already been demonstrated in the ENUBET project [54].

Additionally, a study will be performed on the effect of gadolinium doping of the proposed ESSnuSB water Cherenkov detectors on significance and precision of proposed measurements. Gadolinium has a large cross section for neutron absorption, after which it emits several gamma rays with a total energy of about 8 MeV. This makes it possible to detect neutrons present in the final state of neutrino interactions occurring in a WC detector. Since neutrinos tend to produce protons in the final state, and antineutrinos tend to produce neutrons, by detecting the delayed gamma ray signal from neutron capture one can have a degree of discrimination between neutrino and antineutrino interactions even in the absence of a magnetic field [55,56].

Apart from the main CPV measurement program, the large far detectors of the ESSnuSB project have the potential for additional notable measurements. In the neutrino sector, they will be able to measure interactions of atmospheric neutrinos, solar neutrinos, and supernova neutrinos; additionally, they may be sensitive to neutrino sources that are more difficult to measure, such as cosmic neutrinos, diffuse supernova neutrinos, and geoneutrinos. Given the large water fiducial mass, the far detectors will have a high sensitivity to proton decay as well.

Along with their main purpose of measuring the neutrino interaction cross-section, the proposed neutrino facilities at ESS—LEnuSTORM, LEMNB, and ESSnuSB target station as neutrino sources, in conjuction with ESSnuSB and LEnuSTORM/LEMNB near detectors—will be used for additional short-baseline neutrino physics measurements. One of the main topics of study will be sterile neutrino oscillations driven by a 1 eV$^2$ neutrino mass-square difference. Additional topics will include studies of non-standard neutrino interactions and constraints of new physics scenarios by studying neutrino-electron scattering.

## 7. Conclusions

The ESSnuSB is designed to be a next-to-next generation neutrino oscillation experiment to precisely measure the CP violation phase $\delta_{CP}$ at the second oscillation maximum by employing the uniquely powerful ESS accelerator as a proton driver for neutrino beam production. It is expected to start taking data after HyperK and DUNE have already conducted their measurement and either confirmed the existence of leptonic CPV or excluded it in their sensitivity regions. If the existence of CPV is confirmed, ESSnuSB will start the precision era of leptonic CPV measurement; if not, it will have an equally important mission of searching for CPV in the part of parameter space inaccessible to its two predecessor experiments.

The conceptual design for the ESSnuSB experiment has been published [25,26], and the new project ESSnuSB+ is underway to design intermediate facilities for R&D and measurement of the neutrino interaction cross-section, perform civil engineering conceptual studies, and explore the additional physics possibilities of the proposed infrastructure.

**Funding:** This project has been supported by the COST Action EuroNuNet: "Combining forces for a novel European facility for neutrino-antineutrino symmetry-violation discovery". It has also received funding from the European Union's Horizon 2020 research and innovation programme under grant agreement No. 777419. We acknowledge further support provided by the following research funding agencies: Centre National de la Recherche Scientifique and Institut National de Physique Nucléaire et de Physique des Particules, France; Deutsche Forschungsgemeinschaft, Germany, Projektnummer 423761110; Agencia Estatal de Investigacion through the grants IFT Centro de Excelencia Severo Ochoa, Spain, contract No. CEX2020-001007-S and PID2019-108892RB funded by MCIN/AEI/10.13039/501100011033; Polish Ministry of Science and Higher Education, grant No. W129/H2020/2018, with the science resources for the years 2018–2021 for the realisation of a co-funded project; Ministry of Science and Education of Republic of Croatia grant No. KK.01.1.1.01.0001; Çukurova University Scientific Research Projects Unit, Grant no: FUA-2021-12628; as well as support provided by the universities and laboratories to which the authors of this report are affiliated, see the author list on the first page. Funded by the European Union. Views and opinions expressed are however those of the author(s) only and do not necessarily reflect those of the European Union. Neither the European Union nor the granting authority can be held responsible for them.

**Data Availability Statement:** Not applicable.

**Conflicts of Interest:** The authors declare no conflict of interest.

## Abbreviations

| | |
|---|---|
| CPV | Charge-parity violation |
| ESS | European spallation source |
| ESSnuSB | European spallation source neutrino super beam |
| PMNS | Pontecorvo–Maki–Nakagawa–Sakata |
| NC | Neutral current |
| CC | Charged current |
| QCD | Quantum chromodynamics |
| QES | Quasi-elastic scattering |
| RES | Resonant scattering |
| DIS | Deep-inelastic scattering |
| MEC | Meson-exchange current |
| SFGD | Super fine-grained detector |
| CH | Hydrocarbons |
| PMT | Photomultiplier tube |
| R&D | Research and development |
| LEnuSTORM | Low-energy neutrinos from stored muons |
| LEMNB | Low-energy monitored neutrino beam |
| WC | Water Cherenkov |

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
