# Peer review of "The ESSnuSB Design Study: Overview and Future Prospects"

_universe, doi:10.3390/universe9080347_

Round 1

Reviewer 1 Report

The paper presents a well-written and well-structured summary of oscillation in vacuum (including the discussion of the advantages of the second oscillation maximum) and of the description of the ESSnuSB experiment (apparatus and physics scope).

I have no objection to the publication of the submitted paper and recognize it's a very useful review.

- lines 14-16: It's usual but it's not correct to call the parity transformation as a mirror image (which changes only 2 coordinates). I suggest to improve the text to avoid misunderstanding;

- line 59: it seems the product A_ij is mixing the convention used in Eq. 1 and the one used at PDG. The authors should double check it;

- line 175: there is a typo "... and a very large ..." -- remove "a"

- line 242: there is a typo "... m meter ..."

Author Response

We would like to thank the reviewer for a positive review and the useful comments. The annotated changes to the text are in the attachment and the answers to the comments are below.

Comments on the Quality of English Language

- lines 14-16: It's usual but it's not correct to call the parity transformation as a mirror image (which changes only 2 coordinates). I suggest to improve the text to avoid misunderstanding;
Thank you for this comment, it is indeed a common source of misunderstanding. Added a footnote on L15 with an explanation.

- line 59: it seems the product A_ij is mixing the convention used in Eq. 1 and the one used at PDG. The authors should double check it;
Thank you for spotting this! This was an error while typing the equation and is now corrected.

- line 175: there is a typo "... and a very large ..." -- remove "a"
Thank you, it is now rephrased.

- line 242: there is a typo "... m meter ..."
Corrected.

Reviewer 2 Report

The article  describes  a very ambitious  Project using the European Spallation Source.  It is new to me and I studied it in detail. It is a next-to-next generation project, which means it will be constructed and run after the first experiments aiming to detect CP violation leptonic reactions (after HyperK and DUNE).

      The article gives many details and I am convinced that will succeed. The first experiments will be at the second maximum of the oscillations and will use special desing properties of the beam and the detectors to make them more efficient. At every state of the project I notice innovative ideas and wish them full success. 

Author Response

We would like to thank the reviewer for a positive review. The annotated changes to the text are in the attachment.

Reviewer 3 Report

This is a very good description of the proposed ESSnuSB project, which should be published after a few minor, and mostly grammatical corrections.  I have just four comments or corrections of any scientific substance:

Figure 4: It would be helpful to mention in the text how the polarity flip is achieved.

Line 218: I assume that the SFGD-like detector is made with plastic scintillator.   If that is true then “carbohydrates” should be “hydrocarbons”.  Also on line 406.

Figure 9: I don’t understand why there is no gain in CP coverage between 15 and 17 years.  Is this an artifact, or is there a good reason?

Line 354: Gadolinium neutron capture results in the emission of several gamma rays with a total energy of ~8 MeV.

Line 148: Add “to” Between “transferred” and “a separate”

Line 156: Change “lead” to “led”, and again on line 326.

Figure 4: The caption to figure 4 (a) is partially covered by figure 4 (b)

Lines 175 and 176: The authors have mixed singular and plural forms “a … detectors”. I think they meant to say “two very large water Cherenkov far detectors”

Line 184: Need to add “the” before “far”.  Also on line 194 before “Near”, line 195 before “emulsion”, line 202 before “MEC”, line 208 before “kinematics”, line 210 before “CH”, line 222 before “near”, line 226 before “bulk” and “neutrino”, line 230 before “neutrino”, line 231 before “neutrino”, line 241 before “extreme”, line 252 before “QES”,  figure 7 before “Dashed” and “efficiency”, line 276 before “CPV “ and “expected”, line 298 before “Hyper-K”, line 349 before “long”, line 359 before “absence”, line 365 before “far”, line 367 before “neutrino”, line 370 before “main”, and line 386 before “neutrino”.

Line 198: Change “a 1 ton of” to “a 1 ton” or “1 ton of”

Line 201: Change “scattering like” to “scattering-like”

Line 221: Change “(i) to provide” to “(i) provide”

Line 231: Add “a” before “measurement”

Line 240: Remove “the” before “outward-facing”.  Also on line 296 before “CPV”

Line 297: Before “large”, change “the” to “a”

Line 340: Change “by” to “with”

Author Response

We would like to thank the reviewer for a positive review and the useful comments. The annotated changes to the text are in the attachment and the answers to the comments are below.

Comments and Suggestions for Authors

This is a very good description of the proposed ESSnuSB project, which should be published after a few minor, and mostly grammatical corrections.  I have just four comments or corrections of any scientific substance:

Figure 4: It would be helpful to mention in the text how the polarity flip is achieved.
Indeed, this was not clear from the text, thank you. Removed the word "polarity", explained how to switch between neutrino and antineutrino mode (L157-159).

Line 218: I assume that the SFGD-like detector is made with plastic scintillator.  If that is true then “carbohydrates” should be “hydrocarbons”.  Also on line 406.
Thank you for pointing this out. Changed carbohydrates->hydrocarbons, but left the abbreviation "CH" as it is common in the literature.

Figure 9: I don’t understand why there is no gain in CP coverage between 15 and 17 years.  Is this an artifact, or is there a good reason?
This is a numerical artifact. We expect it to disappear in the next iteration of the analysis. 

Line 354: Gadolinium neutron capture results in the emission of several gamma rays with a total energy of ~8 MeV.
Thank you for the details. Added this info at L358 and two additional references at L363.

Comments on the Quality of English Language

Thank you for these comments. All of them have been implemented.

Line 148: Add “to” Between “transferred” and “a separate”

Line 156: Change “lead” to “led”, and again on line 326.

Figure 4: The caption to figure 4 (a) is partially covered by figure 4 (b)

Lines 175 and 176: The authors have mixed singular and plural forms “a … detectors”. I think they meant to say “two very large water Cherenkov far detectors”

Line 184: Need to add “the” before “far”.  Also on line 194 before “Near”, line 195 before “emulsion”, line 202 before “MEC”, line 208 before “kinematics”, line 210 before “CH”, line 222 before “near”, line 226 before “bulk” and “neutrino”, line 230 before “neutrino”, line 231 before “neutrino”, line 241 before “extreme”, line 252 before “QES”,  figure 7 before “Dashed” and “efficiency”, line 276 before “CPV “ and “expected”, line 298 before “Hyper-K”, line 349 before “long”, line 359 before “absence”, line 365 before “far”, line 367 before “neutrino”, line 370 before “main”, and line 386 before “neutrino”.

Line 198: Change “a 1 ton of” to “a 1 ton” or “1 ton of”

Line 201: Change “scattering like” to “scattering-like”

Line 221: Change “(i) to provide” to “(i) provide”

Line 231: Add “a” before “measurement”

Line 240: Remove “the” before “outward-facing”.  Also on line 296 before “CPV”

Line 297: Before “large”, change “the” to “a”

Line 340: Change “by” to “with”